# Performance of Plasma Phosphorylated tau-217 in Patients on the Continuum of Alzheimer’s Disease

**DOI:** 10.3390/ijms26146771

**Published:** 2025-07-15

**Authors:** Farida Dakterzada, Ricard López-Ortega, Alba Vilella-Figuerola, Nathalia Montero-Castilla, Iolanda Riba-Llena, Maria Ruiz-Julián, Alfonso Arias, Jordi Sarto, Nuria Tahan, Gerard Piñol-Ripoll

**Affiliations:** 1Unitat de Trastorns Cognitius, Cognition and Behaviour Study Group, Hospital Universitari Santa Maria de Lleida, Universitat de Lleida, IRBLleida, 25198 Lleida, Spain; fdakterzada@irblleida.cat (F.D.); avilella@irblleida.cat (A.V.-F.); nmc24@alumnes.udl.cat (N.M.-C.); yriba@gss.cat (I.R.-L.); mruizj@gss.cat (M.R.-J.); aarias@gss.cat (A.A.); jsarto@gss.cat (J.S.); ntahan@gss.cat (N.T.); 2Laboratori Clinical ICS, Hospital Universitari Arnau de Vilanova, 25198 Lleida, Spain; ricardlopez@comll.cat; 3Unitat d’Alzheimer i Altres Trastorns Cognitius, Hospital Clínic de Barcelona–Fundació de Recerca Clínic Barcelona–Institut d’Investigacions Biomèdiques August Pi i Sunyer (FRCB–IDIBAPS), 08036 Barcelona, Spain

**Keywords:** Alzheimer’s disease, plasma, p-tau217, p-tau181, Lumipulse, mild cognitive impairment, biomarker, progression

## Abstract

Recent studies have demonstrated the high analytical and diagnostic performance of plasma p-tau217 using well-defined cohorts. We aimed to assess the analytical, diagnostic, and prognostic utility of plasma p-tau217 as a routine biomarker in symptomatic patients attending our memory clinic. We also sought to identify optimal cutoff points that align with cerebrospinal fluid (CSF) amyloid beta (Aβ) status. A total of 276 cognitively impaired patients were included, with 81 mild cognitive impairment (MCI) patients followed for a mean of 56 (±15.8) months to evaluate progression to Alzheimer’s disease (AD). CSF and blood biomarkers of AD were quantified using the Lumipulse G platform. Plasma p-tau217 levels showed strong correlations with CSF Aβ42/Aβ40 (*r* = −0.707), p-tau181/Aβ42 (*r* = 0.842), and p-tau181 (r *=* 0.728). Plasma p-tau217 levels were significantly higher in the A + T + group than in A − T +/− (*p* < 0.001) and outperformed other plasma markers in detecting CSF Aβ pathology (AUC 0.924).Additionally, p-tau217 moderated cognitive changes over time as measured by the Mini-mental state examination (MMSE) (F(2, 70) = 13.995, *p* < 0.001) and outperformed other plasma biomarkers in predicting progression from MCI to AD (AUC 0.876). Using a dual cutoff strategy, 72% of patients were classified with 94.9% concordance with CSF Aβ status. Plasma p-tau217 shows strong potential as a non-invasive, cost-effective diagnostic and prognostic tool in clinical settings.

## 1. Introduction

Cerebrospinal fluid (CSF) and imaging biomarkers have long been essential tools for accurately diagnosing Alzheimer’s disease (AD) and linking clinical manifestations to underlying pathology [1,2,3,4]. However, their invasiveness and high cost have limited their widespread use in clinical practice, clinical trials, and research studies. As a result, in recent years, there has been a growing effort to measure AD-specific biomarkers in blood as a less invasive and more cost-effective alternative. In this regard, the development of several highly sensitive platforms has helped overcome one of the most significant challenges—the low concentration of AD biomarkers—enabling more accurate quantification in blood [5,6,7,8,9,10,11,12]. Now, kits for determination of plasma amyloid beta 42 (Aβ42), Aβ40, total tau, hyperphosphorylated tau (p-tau), neurofilament light chain (NfL), and Glial fibrillary acidic protein (GFAP) are available for research purposes. Among these biomarkers, hyperphosphorylated tau species (p-tau181, p-tau231, p-tau217) have demonstrated superior diagnostic accuracy and stronger concordance with cerebral and CSF amyloid pathology [5,13,14,15,16,17,18]. In particular, p-tau217 has garnered special attention due to its superior performance compared to other p-tau species in distinguishing patients at different stages of AD, from preclinical to dementia, and differentiating AD from other neurodegenerative disorders [15,19,20,21]. It has shown the largest fold changes in symptomatic AD and better predictive ability for identifying cognitive decline [18,19,22,23].

Based on recent evidence, the Alzheimer’s Association workgroup has updated the diagnostic criteria for AD. According to these revised guidelines, an abnormal result for a core 1 biomarker—such as elevated plasma p-tau217—is now considered sufficient to establish an AD diagnosis [4]. Furthermore, with the approval of disease-modifying monoclonal antibodies and the growing demand for cost-effective, less invasive diagnostic tools, evaluating the performance of p-tau217 and establishing accurate, region-specific cutoff values based on patient characteristics have become essential for memory clinics. Additionally, further research is warranted to assess the utility of this biomarker in monitoring clinical progression in symptomatic individuals on the continuum of AD. By incorporating longitudinal clinical outcomes, our study aims to provide real-world evidence of the long-term prognostic value of p-tau217 in memory clinic patients, highlighting its potential role in risk stratification and guiding early intervention strategies.

Therefore, we aimed to evaluate the clinical and analytical performance of p-tau217 in routine practice among patients seen at our memory clinic. Our objectives were (a) to assess the correlations between plasma p-tau217 and CSF AD biomarkers (Aβ42, Aβ40, total tau, and p-tau181), (b) to determine whether the presence of comorbidities affects plasma levels of this biomarker, (c) to determine whether p-tau217 levels are significantly different between patients with different AD pathology profile, (d) to assess the association between baseline plasma p-tau217 levels and cognitive decline over time, and to determine the long-term prognostic value of p-tau217 in predicting progression from mild cognitive impairment (MCI) to AD, and finally (e) to determine the two cutoff points for p-tau217 based on abnormal Aβ pathology detected in CSF in order to assess the extent to which this biomarker can reduce the number of lumbar punctures necessary for diagnosis of AD in our memory clinic.

## 2. Results

### 2.1. Participants Characteristics

The characteristics of the study population are summarized in Table 1. The average age of the participants was 73 (6.5 SD) years, and females represented 59% of the study population. Significant differences between diagnostic groups were observed for sex (*p* = 0.006), education (*p* = 0.047), Mini-mental state examination (MMSE, *p* < 0.001), and the frequency of the apolipoprotein E ε4 (*APOE* ε4) allele (*p* = 0.013). Among CSF AD biomarkers and ratios, only the levels of Aβ40 did not differ significantly between groups. However, for plasma biomarkers, neither Aβ40 nor the Aβ42/40 ratio showed a significant difference between diagnostic groups (Table 1). We also divided the participants based on their CSF Aβ42/40 status. Of a total of 276 participants, 99 were Aβ− and 177 Aβ+. In this case, the groups were significantly different regarding age (*p* = 0.011), sex (*p* < 0.001), and the frequency of the *APOE* ε4 allele (*p* < 0.001). Among CSF and plasma AD biomarkers and ratios, only the levels of Aβ40 did not differ significantly between groups (Appendix A).

### 2.2. Correlation Between Plasma Levels of p-tau217 and Other Plasma and CSF AD Biomarkers Levels

To evaluate the correlations between CSF and plasma AD biomarkers, we performed Spearman’s rank correlation analysis. The correlation coefficient (*r*) between plasma p-tau217 and plasma p-tau181, CSF Aβ42/40, CSF p-tau181, and CSF p-tau181/Aβ42 were 0.837, −0.707, 0.728, and 0.842, respectively (*p* < 0.001) (Figure 1). The correlations between p-tau217 and all CSF and plasma AD biomarkers are presented in Appendix A. In addition, we examined whether the strength of the correlation between plasma p-tau217 and CSF AD biomarkers, particularly Aβ42/40, differs between male and female participants and between carriers and non-carriers of *APOE* ε4 allele. Our analysis, adjusted for age, *APOE* ε4, and estimated glomerular filtration rate (eGFR), showed that male participants exhibited a slightly stronger correlation between plasma p-tau217 and CSF Aβ42/40 (r = −0.725, *p* < 0.001, N = 94) compared to female participants (r = −0.620, *p* < 0.001, N = 140). Conversely, in female participants, the correlation between plasma p-tau217 and CSF p-tau181 (r = 0.718, *p* < 0.001, N = 140) and t-tau (r = 0.665, *p* < 0.001, N = 140) were stronger than those observed in males (r = 0.641 for p-tau181 and r = 0.498 for t-tau, *p* < 0.001, N = 94) (Appendix A). Furthermore, *APOE* ε4 non-carriers exhibited a stronger correlation between plasma p-tau217 and all CSF AD biomarkers, particularly the Aβ42/40 ratio (r = −0.731, *p* < 0.001, N = 142), compared to *APOE* ε4 carriers (r = −0.466, *p* < 0.001, N = 92) (Appendix A).

### 2.3. Identifying the Factors That May Influence Plasma Levels of p-tau217

Using a multivariate model, we assessed whether demographic data, analytical variables, and common AD comorbidities influence log-transformed plasma p-tau217. Vascular risk factor (VRF) was considered positive if the subject had hypertension, diabetes mellitus, and/or dyslipidemia. The renal function was measured using eGFR. All the presented coefficients seen in Figure 2 were obtained from the model, including all the listed variables. As shown, among the variables examined, CSF Aβ42/40 positivity (β = 0.690, 95% CI 0.3591–0.789, *p* < 0.001) had the largest effect on plasma p-tau217. In addition, reduced renal function was associated with higher levels of p-tau217 (β = −0.113, 95% CI: −0.201 to −0.025, *p* = 0.012). The model had an adjusted R^2^ value of 0.503.

We also evaluated the effect of the same variables on plasma levels of p-tau181 (N = 126). In this case, in addition to CSF Aβ42/40 positivity with the largest effect (β = 0.530, 95% CI 0.385–0.674, *p* < 0.001) and the renal function (β = −0.187, 95% CI: −0.350 to −0.023, *p* = 0.026), the presence of the *APOE* ε4 allele was also related to the plasma levels of p-tau181 (β = −0.250, 95% CI: − 0.406 to −0.095, *p* = 0.002) (Appendix A).

### 2.4. Plasma p-tau217 Levels by Amyloid and Tau Status in CSF

To assess whether plasma p-tau217 levels are associated with AD-related pathology, participants were stratified based on their CSF Aβ42/40 (A) and p-tau181 (T) status, independent of clinical diagnosis. Plasma p-tau217 concentrations differed significantly among the A − T +/− (N = 88), A + T − (N = 23), and A + T + (N = 147) groups (*p* < 0.001) (Figure 3), with the highest levels observed in the A + T + and lowest in the A − T +/− group. Mean (SD) plasma p-tau217 levels were 0.143 (0.143) pg/mL in the A − T +/− group, 0.281 (0.312) pg/mL in the A + T −, and 0.55 (0.398) pg/mL in the A + T+ group.

### 2.5. Accuracy in Discriminating Abnormal Amyloid and Tau Pathologies

To assess the ability of plasma p-tau217 to discriminate CSF Aβ42/40 positive from negative individuals, a binary logistic regression analysis was performed. Our analysis demonstrated that plasma p-tau217 levels have strong discriminative power in distinguishing CSF Aβ42/40 positive from negative individuals (AUC 0.924 95% CI 0.889–0.960), outperforming other plasma biomarkers. p-tau217/Aβ42 ratio yielded a similar classification performance as p-tau217 (Table 2, Figure 4A). In addition, among patients with positive state of CSF Aβ42/40, p-tau217 and p-tau217/Aβ42 showed similar accuracies for abnormal CSF p-tau181 (AUC 0.798, 95% CI 0.701−0.894 and AUC 0.804, 95% CI 0.675−0.933, respectively), which was higher compared with other plasma AD biomarkers (Table 3, Figure 4B).

### 2.6. Relationship Between Plasma Levels of p-tau217 and Longitudinal Changes in Cognition

To evaluate the relationship between plasma levels of AD biomarkers and the cognitive state of our study population, we conducted two analyses. First, we assessed the linear relationship between plasma levels of biomarkers and the baseline MMSE using Spearman’s rank correlation. Our analyses revealed that among the AD plasma biomarkers, only p-tau217 (*r* = −0.314, *p* < 0.001, N = 251) (Figure 5A) and p-tau217/Aβ42 (*r* = −0.200, *p* < 0.038, N = 108) were significantly correlated with MMSE. Based on these findings, we further examined the effect of plasma p-tau217 on cognitive changes, specifically the difference in MMSE between the baseline and the final visit. Our results indicated that plasma p-tau217 significantly moderated pre-post changes in MMSE (2, 70) = 13.995, *p* < 0.001), with higher plasma p-tau217 levels being associated with a more pronounced reduction in MMSE (Figure 5B). Classifying participants into three groups based on plasma p-tau217 level percentiles yielded consistent results. The moderating effect of p-tau217 on changes in the MMSE over time was strongest in participants with high levels of p-tau217 (t (70) = 7.905, *p* < 0.001), followed by those with intermediate levels (t (70) = 2.694, *p* = 0.009). This effect was not observed in participants with low levels of p-tau217 (t (70) = 1.061, *p* = 0.293).

### 2.7. Determining the Predictive Power of Plasma Biomarkers Regarding MCI to AD Progression

After observing that plasma p-tau217 can significantly moderate longitudinal cognitive changes, we evaluated the predictive performance of this biomarker for identifying individuals at risk of progressing from MCI to AD. Among the MCI subjects, 81 were followed up for a mean duration of 4.7 years (SD = 1.3) to assess their clinical progression to dementia. Of these, 45 (55.6%) progressed to dementia (41 to AD, 2 to bvFTD, and 1 to LBD), while 36 (44.4%) subjects did not progress. Our results showed that p-tau217 outperformed other plasma biomarkers in accurately predicting progression from MCI to AD. Additionally, the p-tau217/Aβ42 ratio did not enhance the predictive power of this biomarker (Table 4).

### 2.8. Reference Ranges for Plasma p-tau217 Based on CSF Aβ42/40 Status

We first determined a binary reference point for p-tau217 based on CSF Aβ42/40 positivity. In this case, the reference point ≥0.237 pg/mL for p-tau217 was the value that yielded the maximum Youden index versus CSF Aβ42/40 status. The overall percent agreement OPA was 87.9% (Table 5). At the next step, we applied a three-range approach to determine the lower (95% or 97.5% sensitivity, p-tau217 < 0.156 pg/mL or <0.1285 pg/mL, respectively) and upper (95% or 97.5% specificity, ≥0.385 pg/mL or ≥0.841 pg/mL, respectively) reference points for p-tau217 (Table 5, Figure 6). This approach improved the OPA for p-tau217 (Table 5).

## 3. Discussion

In this study, we evaluated the diagnostic and prognostic performance of plasma AD biomarkers in a memory clinic cohort of patients with MCI, AD, and non-AD dementia using commercially available Lumipulse G assays. Our results indicated that plasma p-tau217 has a high correlation with CSF Aβ42/40, p-tau181, p-tau181/Aβ42, and t-tau/Aβ42 and plasma p-tau181. Among plasma biomarkers, p-tau217, p-tau181, and their ratio with Aβ42 demonstrated high accuracy in discriminating patients with abnormal amyloid pathology in CSF. However, p-tau217 outperformed other plasma biomarkers in accurately distinguishing the presence of tau pathology in CSF among Aβ-positive individuals. Additionally, among plasma AD biomarkers, only p-tau217 significantly correlated with the measurement of cognition by MMSE. The higher levels of this biomarker in plasma were associated with a more pronounced reduction in MMSE over time. p-tau217 also had a better performance in predicting MCI to AD progression compared to other plasma biomarkers. In all analyses, the performance of p-tau217/Aβ42 was equivalent, but not superior to p-tau217. A lenient three-range reference points (95% sensitivity and specificity) indicated that the use of plasma p-tau217 as a diagnostic biomarker in our memory clinic can reduce the number of lumbar punctures by nearly 72%, while applying more restrictive cutoffs (97.5% sensitivity and specificity) reduces this number to 40%.

The performance of plasma biomarkers quantified using different analytical platforms to detect AD pathology has been assessed in previous studies [18,24,25,26,27,28]. Nearly all studies consistently report that plasma p-tau217 outperforms other plasma markers in detecting amyloid brain pathology, whether identified via CSF analysis or PET imaging. In line with these previous findings, our analysis also showed that plasma p-tau217 surpassed other plasma biomarkers in detecting amyloid pathology in CSF. Importantly, plasma p-tau217 was not outperformed by p-tau217/Aβ42, further supporting its potential as a reliable single biomarker.

Consistent with previous findings [18,21], we observed that among Aβ-positive individuals, the p-tau217 assay demonstrated greater accuracy in detecting tau pathology compared to other plasma biomarkers. This is especially important because anti-Aβ treatments may have limited efficacy in patients with advanced tau pathology [29]. Therefore, the p-tau217 assay could help identify patients who are more likely to benefit from such treatments.

The extent to which plasma p-tau217 concentrations could be affected by subject health condition, genetic, or demographic data is a critical issue, as it could lead to misdiagnosis. For instance, some previous studies have shown that chronic kidney disease can affect plasma p-tau217 levels and have found negative associations between eGFR and this biomarker [30,31]. For this reason, we evaluated the effect size of Aβ positivity on plasma p-tau217 in the presence of common AD comorbidities (VRF and depression), *APOE* ε4, body mass index (BMI), and eGFR. Our analysis showed that, when accounting for all these variables, Aβ positivity had the largest effect on plasma p-tau217 (β = 0.690). In addition, eGFR had a significant negative effect on p-tau217 (β = −0.113), although its impact was much smaller than that of Aβ positivity, consistent with findings from previous studies [25,26]. Importantly, prior evidence indicates that chronic kidney disease does not affect the accuracy of plasma p-tau217 in detecting abnormal AD pathology identified in CSF [32]. Nevertheless, this association should be carefully considered, particularly in patients with severe kidney disease.

In contrast, for p-tau181, the variables with significant effect were Aβ positivity, *APOE* ε4, and eGFR. Specifically, the presence of the *APOE* ε4 allele was associated with lower plasma p-tau181 levels. While some previous studies in cognitively healthy older adults have reported higher plasma p-tau181 in *APOE* ε4 carriers compared to non-*APOE* ε4 carriers [33], no such difference was observed between *APOE* ε4 carriers and non-carriers among patients on the AD continuum [34,35]. Therefore, these discrepancies in the relationship between APOE ε4 and p-tau181 may be partly attributable to differences in the study populations and the degree of cognitive impairment.

In addition to determining the diagnostic utility, we also assessed the prognostic performance of plasma p-tau217 in predicting MCI to AD progression and the rate of cognitive decline. Plasma p-tau217 outperformed other plasma biomarkers in predicting MCI to AD progression. Among the plasma biomarkers, only p-tau217 and p-tau217/Aβ42 showed a significant correlation with baseline cognitive status as measured by the MMSE, with the correlation being stronger for p-tau217 than for the ratio. Additionally, plasma p-tau217 moderated longitudinal changes in cognition, as the participants with higher p-tau217 levels experienced a faster rate of cognitive decline compared to those with lower levels of this biomarker. Consistent with our observations, some earlier studies have also demonstrated the utility of plasma p-tau217 in predicting cognitive decline [17,19,28] and conversion to AD dementia [36].

The prevalence of Aβ pathology varies according to age, severity of clinical symptoms, race and/or ethnicity, sex, and *APOE* genotype [37]. Consequently, blood concentrations of p-tau217, as the best blood biomarker of brain Aβ pathology, are likely influenced by these factors. Therefore, it is reasonable for each center to establish its own cutoffs based on the characteristics of its cognitively impaired population.

In this study, we first determined a binary cutoff for p-tau217 to assess its accuracy in detecting CSF Aβ pathology, based on recently published criteria requiring a minimum accuracy of ≥90% [37]. Our analysis showed that a single cutoff for p-tau217 (≥0.2375 pg/mL, accuracy = 87.9%) did not meet this threshold. Given the limited accuracy of a single reference point—largely because 5–20% of individuals present borderline Aβ pathology—it is highly recommended to use two cutoffs to define three categories of results: positive, intermediate, and negative. We tested two three-range strategies: a lenient one with 95% sensitivity and specificity, and a restrictive one with 97.5% sensitivity and specificity. The lenient cutoffs yielded better results, achieving high positive and negative percent agreements (PPA and NPA) of 95.7% and 88.4%, respectively, with only 27% of individuals falling into the intermediate-risk group requiring confirmatory or follow-up testing. In contrast, the restrictive strategy placed about 61% of the study population in the intermediate zone.

Of note, our higher reference point (0.385 pg/mL) closely matched values reported in previous studies conducted in Spain (0.388 pg/mL in [26]; and 0.354 pg/mL in [25]) involving memory clinic cohorts. However, our lower reference point (0.156 pg/mL) differed somewhat from those studies (0.186 pg/mL in [26]); and (0.185 pg/mL in [25]). This discrepancy may stem from differences in the studied cohort characteristics, which, as mentioned earlier, can influence brain Aβ pathology levels and, consequently, plasma concentrations of p-tau217.

Our study has several strengths. First, we detected all CSF and plasma biomarkers using commercially available automated Lumipulse G assays. This platform is already present in many clinical settings, and therefore, our results may facilitate the clinical implementation of this biomarker. Second, the personnel who quantified the biomarkers and analyzed the data were blinded to the clinical information. Third, since the clinical use of blood biomarkers is currently intended for the evaluation of symptomatic individuals rather than cognitively unimpaired ones, we included only individuals with objective memory impairment in our study. Consequently, our study population is representative of those who would require a blood test to evaluate the underlying cause of their cognitive deterioration in a memory clinic. However, our study has some limitations. The sample size was small, and blood biomarkers other than p-tau217 were not quantified for the entire study population. Additionally, the lack of pathophysiological confirmation or Aβ/tau PET imaging in our cohort may have affected our results.

Taken together, our results support the inclusion of plasma p-tau217 in the current diagnostic workup to reduce the need for more costly, invasive, and less accessible confirmatory tests, such as CSF analysis and PET imaging. Additionally, plasma p-tau217 shows promise as a prognostic indicator for future cognitive decline, supporting its role in patient monitoring and management.

## 4. Material and Methods

### 4.1. Study Population

A total of 276 patients, including 193 MCI, 56 AD, and 27 non-AD dementia patients, were included in this study. The non-AD dementia group included four vascular dementia, ten frontotemporal dementia, two Lewy body dementia, three mixed dementia, one tauopathy, two pharmacological Parkinsonism, one alcoholic dementia, and four unspecified dementia patients. The study population consisted of patients attending the Cognitive Disorders Unit at the Hospital Universitari Santa Maria (Lleida, Spain). The diagnosis of MCI or AD was performed based on the National Institute on Aging–Alzheimer’s Association (NIA-AA) criteria [38,39]. Each non-AD dementia patient fulfilled the specific diagnostic criteria for the disorder considered (e.g., frontotemporal dementia, Lewy body dementia, etc.) [40,41,42]. The inclusion criterion was the presentation of suspected cognitive dysfunction for which the neurologist at the memory unit requested CSF analysis. Therefore, patients with cognitive impairment caused by psychiatric problems or other conditions, such as stroke, brain tumor, and vitamin deficiency were excluded. Epidemiological data, including age, sex, family history of cognitive impairment, education, and comorbidities such as hypertension, diabetes mellitus, dyslipidemia, and depression, were collected through a structured interview conducted during the initial patient visit. Data on eGFR and BMI were also collected and included in the analysis. The included patients or their legal representatives signed an internal regulatory document stating that residual samples used for diagnostic procedures could be used for research studies without any additional informed consent. The study was conducted in accordance with the Declaration of Helsinki.

### 4.2. Sample Collection and Storage

CSF and plasma samples were collected between 8 a.m. and 10 a.m. after an overnight fast. CSF was collected in 10 mL polypropylene tubes (Sarstedt, Newton, NC, USA, 62.610.201). The tubes were inverted several times and centrifuged at 2000× *g* for 10 min at room temperature. The samples were aliquoted into two 2 mL polypropylene tubes (Sarstedt, 72.694.007), with each tube containing 1 mL of CSF. Blood samples were collected in EDTA-containing vacutainer tubes and centrifuged at 2000× *g* for 10 min at 4 °C to separate plasma and the buffy coat. All samples were aliquoted and immediately stored at −80 °C until use. Samples were obtained with support from IRBLleida Biobank (B.0000682) and Biobank and Biomodels Platform ISCIII PT23/00032.

### 4.3. Sample Analysis

The Lumipulse G600II automated platform (Fujirebio Europe NV, Gent, Brussels) was used to measure AD biomarkers in CSF and plasma. CSF biomarkers included Aβ42 (#230336), Aβ40 (#231524), p-tau181 (#230350), and total tau (#230312), while plasma biomarkers comprised Aβ42 (#81301), Aβ40 (#81298), p-tau181 (#81288), and p-tau217 (#81472). The following cut-offs for CSF biomarkers were determined by Fujirebio and used for data analysis: Aβ42 < 600 pg/mL, Aβ42/40 < 0.069, p-tau181 > 56.5 pg/mL, and t-tau > 400 pg/mL. The quantification ranges of plasma Aβ40, Aβ42, p-tau181, and p-tau217 were 0.44–5000 pg/mL, 0.43–1000 pg/mL, 0.261–60 pg/mL, and 0.03–10 pg/mL, respectively. The limit of detection (LOD) was 0.44 pg/mL for plasma Aβ40, 0.37 pg/mL for Aβ42, 0.052 pg/mL for plasma p-tau181, and 0.02 pg/mL for plasma p-tau217. The investigators involved in the sample analyses were blinded to the clinical diagnosis.

### 4.4. Statistical Analysis

Kruskal–Wallis and chi-square tests were used for analysis of quantitative and qualitative variables, respectively. The quantitative variables are presented as medians (25th percentile; 75th percentile), and the qualitative variables are presented as frequencies (percentage). To evaluate the correlation between plasma and CSF biomarkers, we used Spearman’s correlation coefficient (r). Generalized linear mixed model was employed to assess the effect of potential confounding variables on p-tau biomarkers. The diagnostic accuracy of each plasma biomarker for CSF Aβ42/40 positivity was evaluated using binary logistic regression analysis. For each biomarker, we calculated the area under the curve (AUC), sensitivity (percentage of correctly classified positive cases), specificity (percentage of correctly classified negative cases), positive percent agreement (PPA: correctly classified positive cases/[correctly classified positive cases + incorrectly classified negative cases]), negative percent agreement (NPA: correctly classified negative cases/[correctly classified negative cases + incorrectly classified positive cases]), and overall diagnostic accuracy (correctly classified positive and negative cases/total number of cases). The same diagnostic accuracy parameters were also calculated for each plasma biomarker in detecting CSF p-tau181 positivity among Aβ-positive individuals. For evaluating the effect of p-tau217 on cognitive deterioration over time, we applied a general linear mixed model with repeated measures in which the follow-up time was included as a covariate. To determine optimal cutoff values for plasma p-tau217, receiver operating characteristic (ROC) analyses were performed. First, a binary cutoff point for CSF Aβ42/40 positivity was determined using the Youden Index (sensitivity + specificity − 1). Additionally, two three-range reference thresholds were established: a lower cutoff to rule out AD (based on 95% [lenient] and/or 97.5% [restrictive] sensitivity), and an upper cutoff to rule in AD (based on 95% [lenient] and/or 97.5% [restrictive] specificity). All statistical analyses were conducted using SPSS software, version 25 (Armonk, NY, USA).

## Figures and Tables

**Figure 1 ijms-26-06771-f001:**
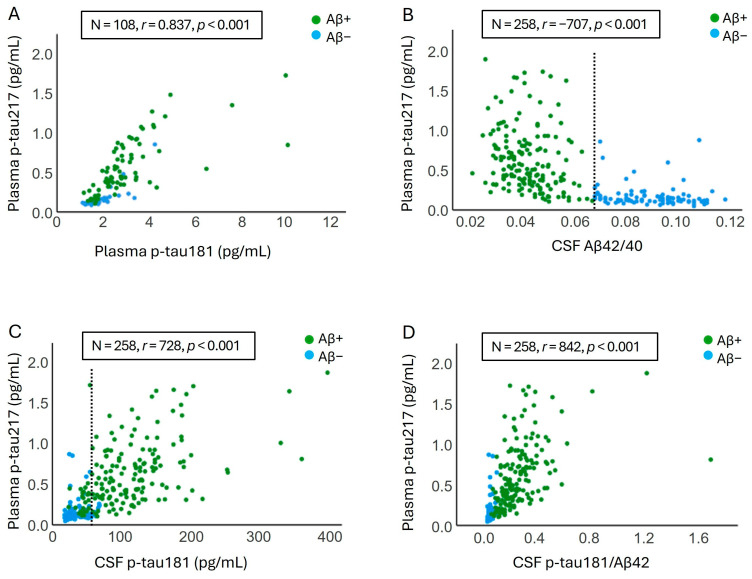
Spearman correlation between plasma p-tau217 levels and other AD-related biomarkers. The correlation between plasma p-tau217 and plasma p-tau181 (**A**), CSF Aβ42/40 (**B**), CSF p-tau181 (**C**), and CSF p-tau181/Aβ42 (**D**) is shown. The dotted line represents the cutoff value for the corresponding CSF biomarker, where available.

**Figure 2 ijms-26-06771-f002:**
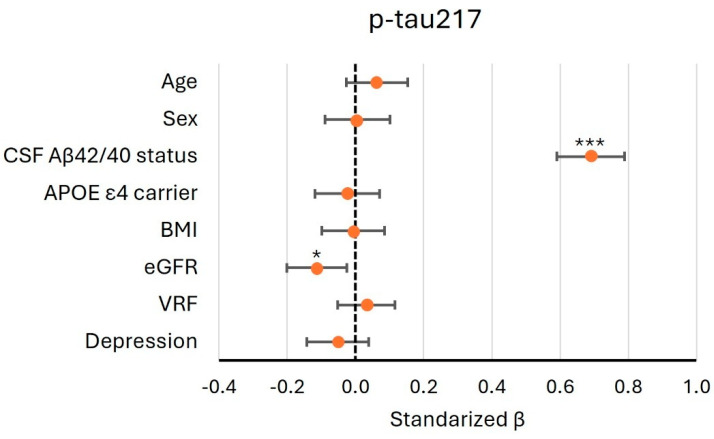
Effect of potential confounding variables on plasma p-tau217. The black vertical dashed line indicates a null effect. Orange dots represent the standardized beta coefficients for each variable, and the horizontal bars indicate the corresponding 95% confidence intervals. BMI: body mass index; eGFR: estimated glomerular filtration rate; VRF: vascular risk factor; *: *p* < 0.05; ***: *p* < 0.001.

**Figure 3 ijms-26-06771-f003:**
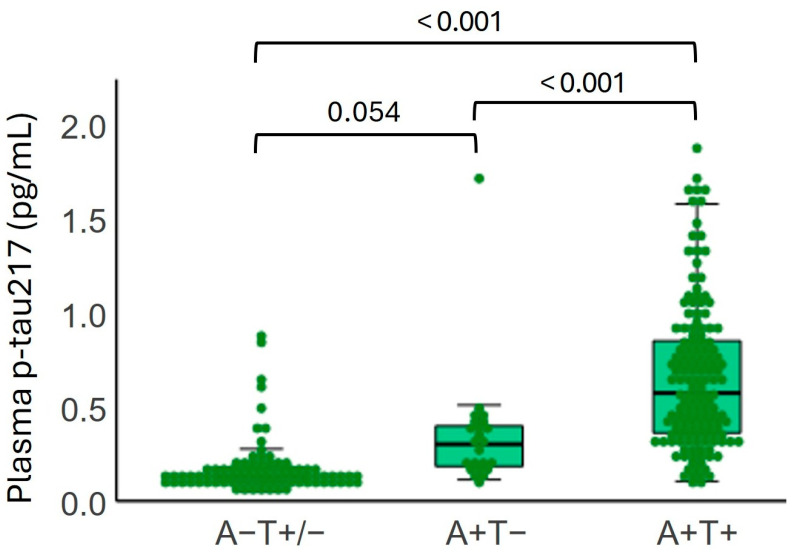
Plasma p-tau217 levels according to CSF Aβ42/40 (A) and p-tau181 (T) status.

**Figure 4 ijms-26-06771-f004:**
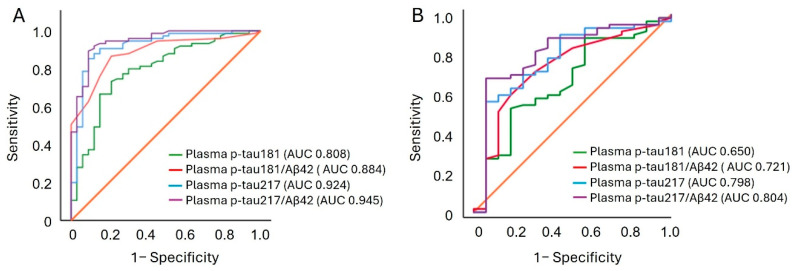
Receiver operating characteristic (ROC) curves of plasma p-tau biomarkers and ratios for discriminating amyloid and tau pathologies in CSF. (**A**) Plasma biomarkers that achieved the highest Youden index for discriminating CSF Aβ42/40 status in the ROC analysis. (**B**) Plasma biomarkers that achieved the highest Youden index for discriminating CSF p-tau181 status in the ROC analysis.

**Figure 5 ijms-26-06771-f005:**
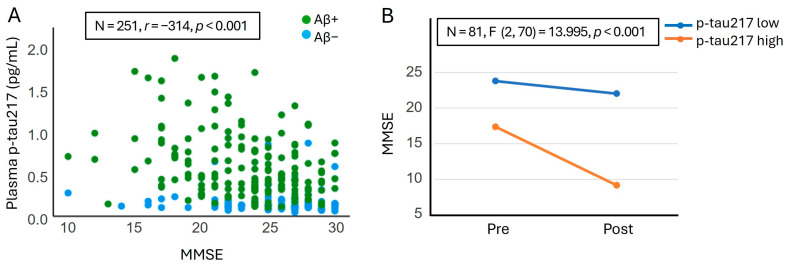
Association between plasma p-tau217 levels and cognition, as measured by the MMSE. (**A**) Spearman correlation between plasma p-tau217 levels and MMSE scores at baseline. (**B**) Moderating effect of plasma p-tau217 on cognitive changes over time in a subpopulation of MCI patients with available follow-up data (N = 81). “Pre” refers to the MMSE score at the baseline visit, and “Post” refers to the MMSE score at the final visit. High p-tau217 levels were defined as the mean plus one standard deviation (mean + SD), and low p-tau217 levels as the mean minus one standard deviation (mean − SD). MMSE: Mini-mental state examination.

**Figure 6 ijms-26-06771-f006:**
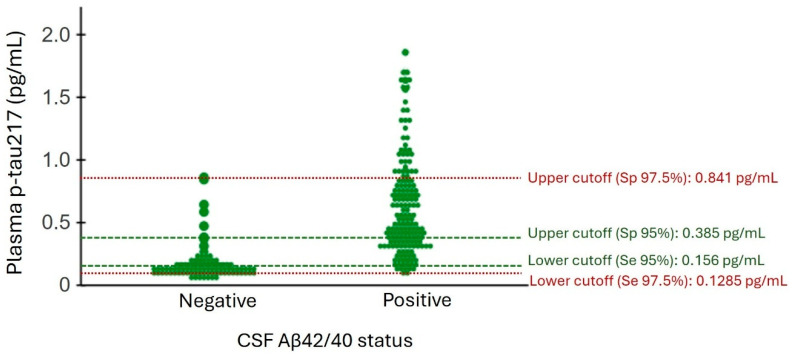
Three-range restrictive and lenient reference points for CSF Aβ status. The red dashed lines indicate the upper and lower reference points for the restrictive strategy, where the specificity of the upper cutoff and the sensitivity of the lower cutoff are both set at 97.5%. Individuals with values between these two red dashed lines (N = 170 (61.6%)) are considered to have an intermediate risk for CSF Aβ positivity and will require confirmatory testing. The green dashed lines represent the upper and lower reference points for the lenient strategy, where the specificity of the upper cutoff and the sensitivity of the lower cutoff are set at 95%. Individuals falling within this range (N = 72 (27.9%)) are also classified as having intermediate risk and will need confirmatory testing for CSF Aβ positivity. Se: sensitivity; Sp: specificity.

**Table 1 ijms-26-06771-t001:** Characteristics of study population by clinical diagnosis.

	N	All Participants	MCI	AD	Non-AD Dementia	*p*-Value
n (%)		276 (100%)	193 (69.9%)	56 (20.3%)	27 (9.8%)	
Demographic data						
Age (years)	276	74 (70, 78)	75 (72, 78)	74 (70, 77)	75 (70, 79)	0.242
Sex (female), n(%)	276	164 (59%)	116 (60%)	39 (70%)	9 (33%)	0.006
Education (years)	264	9 (8, 12)	9 (8, 14)	10 (8, 12)	8 (6, 10)	0.047
Family history of cognitive impairment (yes), n (%)	274	113 (41%)	24 (43%)	80 (42%)	9 (33%)	0.674
Comorbidities						
Hypertension, n(%)	276	153 (55%)	100 (52%)	39 (70%)	14 (52%)	0.057
Diabetes Mellitus, n(%)	276	84 (30%)	60 (31%)	17 (30%)	7 (26%)	0.861
Dyslipidemia, n(%)	276	148 (54%)	107 (55%)	29 (52%)	12 (44%)	0.536
Depression, n(%)	275	98 (35%)	125 (65%)	35 (63%)	17 (63%)	0.926
CSF AD biomarkers						
Aβ42 pg/mL	276	546 (410, 799)	595 (432, 866)	424 (346, 537)	784 (508, 988)	<0.001
Aβ40 pg/mL	276	10788 (8335, 13467)	11121 (8565, 13659)	10246 (8162, 12222)	9215 (8092, 13469)	0.139
Aβ42/40	276	0.051 (0.04, 0.082)	0.052 (0.040, 0.085)	0.041 (0.035, 0.050)	0.082 (0.063, 0.101)	<0.001
t-tau pg/mL	276	455 (268, 747)	425 (258, 661)	653 (432, 897)	265 (170, 487)	<0.001
p-tau181 pg/mL	276	69 (41, 121)	65 (41, 118)	107 (71, 149)	40 (27, 56)	<0.001
t-tau/Aβ42	276	0.9 (0.3, 1.57)	0.79 (0.29, 1.39)	1.60 (0.99, 1.98)	0.29 (0.22, 0.90)	<0.001
p-tau181/Aβ42	276	0.15 (0.04, 0.27)	0.13 (0.04, 0.24)	0.27 (0.15, 0.32)	0.04 (0.03, 0.10)	<0.001
Plasma AD biomarkers						
Aβ42 pg/mL	124	24 (21, 27.8)	24.1 (21.1, 27.2)	22.5 (20.3, 26.6)	25.9 (24.1, 29.2)	0.040
Aβ40 pg/mL	123	303 (276, 359)	309 (276, 359)	283 (272, 334)	320 (282, 371)	0.211
Aβ42/40	123	0.08 (0.07, 0.08)	0.08 (0.07, 0.08)	0.08 (0.07, 0.08)	0.08 (0.07, 0.09)	0.093
p-tau181 pg/mL	126	2.29 (1.69, 3.13)	2.17 (1.65, 2.85)	2.86 (2.28, 3.49)	2.10 (1.62, 3.08)	0.009
p-tau217 pg/mL	258	0.34 (0.15, 0.69)	0.31 (0.14, 0.59)	0.70 (0.39, 0.98)	0.13 (0.11, 0.30)	<0.001
p-tau181/Aβ42	126	0.1 (0.07, 0.14)	0.09 (0.07, 0.12)	0.13 (0.10, 0.15)	0.07 (0.06, 0.13)	0.001
p-tau217/Aβ42	123	0.015 (0.006, 0.028)	0.014 (0.005, 0.023)	0.029 (0.015, 0.038)	0.005 (0.004, 0.013)	<0.001
Other variables						
BMI	258	26.6 (23.89, 29.27)	26.6 (24.3, 29.1)	25.5 (23.4, 30.4)	27.0 (24.6, 28.9)	0.718
eGFR	264	75 (63, 86)	73 (62, 86)	73 (55, 84)	82 (66, 88)	0.349
MMSE score	269	25 (21, 27)	25 (23, 27)	21 (18, 23)	25 (18, 27)	<0.001
*APOE* ε4, n(%)	272	106 (38%)	72 (38%)	29 (52%)	5 (19%)	0.013

Unless otherwise specified, results are presented as median (IQR). *p*-values were calculated by comparing AD, MCI, and non-AD dementia participants using Kruskal Wallis test for continuous variables and Pearson Chi2 for categorical variables. AD: Alzheimer’s disease; *APOE* ε4: apolipoprotein E ε4 allele; BMI: body mass index; eGFR: estimated glomerular filtration rate; MCI: mild cognitive impairment; MMSE: Mini-mental state examination; N: number of included participants; non-AD dementia: non-Alzheimer’s disease dementia.

**Table 2 ijms-26-06771-t002:** Discriminating power of plasma biomarkers for CSF Aβ42/40 status.

Biomarker	AUC (95% CI)	Sensitivity	Specificity	PPA	NPA	Total% of Predictive Accuracy *
Aβ42	0.668 (0.568–0.768)	62.4%	61.5%	77.9%	42.9%	62.1%
Aβ40	0.505 (0.391–0.618)	52.4%	48.7%	69.8%	32.8%	51.6%
Aβ42/40	0.796 (0.713–0.879)	57.1%	89.7%	92.6%	50.0%	68.3%
p-tau181	0.808 (0.727–0.890)	80.2%	75.0%	87.3%	63.8%	78.6%
p-tau181/Aβ42	0.884 (0.824–0.944)	87.1%	82.1%	91.5%	75.0%	85.7%
p-tau217	0.924 (0.889–0.960)	87.3%	89.1%	93.5%	79.6%	87.9%
p-tau217/Aβ42	0.945 (0.900–0.990)	89.3%	90.9%	95.7%	78.9%	89.8%

AUC: area under the curve; NPA: negative percent agreement; PPA: positive percent agreement. * The percentage of correct classification of Aβ42/40 positive + correct classification of Aβ42/40 negative/all cases.

**Table 3 ijms-26-06771-t003:** Discriminating power of plasma biomarkers for CSF p-tau181 status.

Biomarker	AUC (95% CI)	Sensitivity	Specificity	PPA	NPA	Total% of Predictive Accuracy *
Aβ42	0.412 (0.277–0.548)	81.3%	18.8%	75.9%	9.7%	51.7%
Aβ40	0.396 (0.257–0.535)	66.2%	18.8%	77.6%	11.5%	57.1%
Aβ42/40	0.506 (0.355–0.658)	44.1%	62.5%	83.3%	20.8%	47.6%
p-tau181	0.650 (0.498–0.802)	51.4%	81.3%	92.3%	29.2%	57.4%
p-tau181/Aβ42	0.721 (0.580–0.861)	56.5%	81.3%	92.9%	30.2%	61.2%
p-tau217	0.798 (0.701–0.894)	71.3%	78.3%	91.0%	47.6%	85.5%
p-tau217/Aβ42	0.804 (0.675–0.933)	72.9%	75.0%	91.5%	41.4%	72.3%

AUC: area under the curve; NPA: negative percent agreement; PPA: positive percent agreement. * The percentage of correct classification of p-tau181 positive + correct classification of p-tau181 negative/all cases.

**Table 4 ijms-26-06771-t004:** Predictive power of plasma biomarkers for progression from MCI to AD.

Biomarker	AUC (95% CI)	Sensitivity	Specificity
Aβ42	0.374 (0.249–0.500)	46.5%	31.4%
Aβ40	0.460 (0.329–0.591)	41.9%	42.9%
Aβ42/40	0.673 (0.553–0.793)	67.4%	65.7%
p-tau181	0.795 (0.692–0.898)	90.1%	63.9%
p-tau181/Aβ42	0.852 (0.762–0.943)	86.0%	77.1%
p-tau217	0.876 (0.787–0.964)	87.2%	83.9%
p-tau217/Aβ42	0.888 (0.803–0.972)	84.6%	87.1%

**Table 5 ijms-26-06771-t005:** Binary and three-range reference for CSF Aβ42/40 status.

Binary Reference Point ^$^		Three-Range Reference Points		
			95% *	97.5% **
Characteristics	≥0.2375 pg/mL	Characteristics	<0.1560 ≥0.3850 pg/mL	<0.1285 ≥0.8410 pg/mL
Participants, n	258	Participants, n	258	258
Positive for Aβ42/40, n (%)	177 (68.6)	Positive for Aβ42/40, n (%)	177 (68.6)	177 (68.6)
Positive for plasma p-tau217, n (%)	155 (60.1)	Negative for plasma p-tau217, n (%)	69 (26.7)	48 (18.6)
		Intermediate for plasma p-tau217, n (%)	72 (27.9)	170 (61.6)
		Positive for plasma p-tau217, n (%)	117 (45.3)	40 (15.5)
Sensitivity, %	87.3%	Sensitivity of lower reference point, %	95.2%	97.6%
Specificity, %	89.1%	Specificity of upper reference point, %	94.6%	97.8%
PPA, %	93.5%	PPA, upper reference point, %	95.7%	95.0%
NPA, %	79.6%	NPA, lower reference point, %	88.4%	91.7%
OPA, %	87.9%	OPA for plasma p-tau217 positive and negative, %	94.9%	97.7%

$: The value that yielded the maximum Youden index. *: Three-range reference points with 95% sensitivity at the lower reference points and 95% specificity at the upper reference point; **: Three-range reference points with 97.5% sensitivity at the lower reference point and 97.5% specificity at the upper reference point; Aβ42/40: amyloid beta 42 to 40 ratio; CSF: cerebrospinal fluid; NPA: negative percent agreement; OPA: overall percent agreement; PPA: positive percent agreement; p-tau: phosphorylated tau.

## Data Availability

The data reported in this manuscript are available within the article and/or its Appendix A. Additional data will be shared upon request by any qualified investigator.

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
