# Peer review of "Performance of Plasma Phosphorylated tau-217 in Patients on the Continuum of Alzheimer’s Disease"

_ijms, 2025, doi:10.3390/ijms26146771_

Round 1

Reviewer 1 Report

Comments and Suggestions for Authors
  1. Gender and APOE gene types are key factors for AD studies. For the AD p-Tau and other biomarker correlations in CSF and plasma, did authors find any differences between female and male, or different APOE subtypes?
  2. What is the real meaning of p-values in Table 1?
  3. Lines 248-249, the writing format of this sentence is not consistent with the whole manuscript. The same for lines 277-286 Please double check and revise accordingly.
  4. Based on authors summary, the diagnostic accuracy of plasma p-Tau217 outstands the previous CSF analysis and PET imaging based on the cohort, will authors test bigger cohorts to validate these results?
  5. When authors used the plasma p-Tau217 as AD diagnostic marker, authors focused on the average 75 years old group. Nowadays the younger groups in the 40s, 50s, 60s also get ADs. Will p-Tau217 be accurate for the younger groups?

Author Response

The authors sincerely appreciate the reviewers’ time and effort in evaluating our manuscript. Their constructive comments and suggestions have significantly improved the quality of our work. Below, we provide a point-by-point response to all the questions and comments raised by the reviewers.

Reviewer 1:

  1. Gender and APOE gene types are key factors for AD studies. For the AD p-Tau and other biomarker correlations in CSF and plasma, did authors find any differences between female and male, or different APOE subtypes?

R/ Thank you very much for this valuable comment. We conducted correlation analyses between plasma p-tau217 and CSF AD biomarkers splitting the study population once by sex (controlling for age, APOE4, and eGFR) and another time by APOE4 status (controlling for age, sex, and eGFR).  

Our analysis, showed that male participants exhibited a slightly stronger correlation between plasma p-tau217 and CSF Aβ42/40 (r = −0.725, p<0.001, N=94) compared to female participants (r = −0.620, p<0.001, N=140). Conversely, in female participants, the correlation between plasma p-tau217 and CSF p-tau181 (r = 0.718, p<0.001, N=140) and t-tau (r = 0.665, p<0.001, N=140) were stronger than those observed in males (r = 0.641 for p-tau181 and r = 0.498 for t-tau, p<0.001, N=94). Furthermore, APOE ε4 non-carriers exhibited a stronger correlation between plasma p-tau217 and all CSF AD biomarkers, particularly Aβ42/40 ratio (r = −0.731, p<0.001, N=142), compared to APOE ε4 carriers (r = −0.466, p<0.001, N=92), while controlling for age, sex and eGFR. This information was added to the manuscript (lines 170-181) and two supplementary tables (S Table 3 and 4) were also provided.

2. What is the real meaning of p-values in Table 1?

R/ p-values in Table 1 were calculated by comparing participants with AD, MCI, and non-AD dementia using the Kruskal-Wallis test for continuous variables and the Pearson Chi-square test for categorical variables. Although this explanation is included in the footnote of Table 1, the font size may have been too small to read easily. We have now included a version of Table 1 in a larger format for improved readability.

3. Lines 248-249, the writing format of this sentence is not consistent with the whole manuscript. The same for lines 277-286 Please double check and revise accordingly.

R/ The format was adjusted according to the rest of the manuscript.

4. Based on authors summary, the diagnostic accuracy of plasma p-Tau217 outstands the previous CSF analysis and PET imaging based on the cohort, will authors test bigger cohorts to validate these results?

R/ We appreciate the reviewer’s comment. As the reviewer noted, the original conclusion may have implied that the inclusion of plasma p-tau217 improves the diagnostic accuracy of currently used tests, such as CSF analysis and PET imaging. However, we acknowledge that the diagnostic performance of plasma p-tau217 is not superior to CSF or PET, as these modalities are often used as reference standards to evaluate the performance of plasma biomarkers in most studies.

In our study, we assessed the diagnostic performance of plasma p-tau217 using the CSF Aβ42/40 ratio as the reference biomarker for AD-related amyloid pathology. We have revised the conclusion to better reflect the intended message: “Taken together, our results support the inclusion of plasma p-tau217 in the current diagnostic workup to reduce the need for more costly, invasive, and less accessible confirmatory tests, such as CSF analysis and PET imaging.” (Lines 395-397)

5. When authors used the plasma p-Tau217 as AD diagnostic marker, authors focused on the average 75 years old group. Nowadays the younger groups in the 40s, 50s, 60s also get ADs. Will p-Tau217 be accurate for the younger groups?

R/ Thank you for the comment. As the reviewer noted, the average age of our study population was 73 years. However, nearly 20% (19.56%) of the participants were younger than 70 years, ranging from 42 to 69 years old. We did not evaluate the performance of p-tau217 exclusively in younger groups, as most of the visitors of our memory clinics are in their 70s. However, previous studies in preclinical Alzheimer’s disease have demonstrated the utility of this biomarker in detecting brain amyloid pathology in younger populations (e.g. PMID: 37013174 and PMID: 36745413). Furthermore, studies involving individuals with Down syndrome, typically in their 40s, have also shown that plasma p-tau217 accurately identifies individuals with abnormal tau-PET and Aβ-PET scans (PMID: 35789365).

These findings support the accuracy of p-tau217 in younger populations. However, the optimal cutoff values may differ from those established in older populations, as the intended use population significantly influences the cutoff due to differences in brain amyloid load across age groups and the pathology initiation time. However, we do not have a sufficiently large cohort of young individuals to perform statistical analyses with reliable results.

Reviewer 2 Report

Comments and Suggestions for Authors

This manuscript presents a comprehensive evaluation of plasma p-tau217 as a diagnostic and prognostic biomarker for Alzheimer’s disease (AD) in a memory clinic setting. The study is well-conducted. The findings align with current efforts to implement blood-based biomarkers in routine clinical practice and support the strong clinical utility of p-tau217. However, some areas require clarification, improved organization, or further discussion to strengthen the manuscript.

  1. The novelty of the study should be addressed. Specifically, how does this work add to or differ from prior studies?
  2. More detail is needed on the clinical criteria used to diagnose non-AD dementias. For example, please clarify whether a cognitively normal control group was considered or used for reference.
  3. The association between eGFR and plasma p-tau217 is notable. The authors may consider discussing whether adjusting p-tau217 for renal function could further enhance diagnostic accuracy or aid clinical interpretation.
  4. The abstract could be improved by including numerical values (e.g., AUC) when describing the performance of p-tau217.
  5. Too many tables in the main text. The authors may consider moving some tables to supporting information.
  6. Please clarify the cut-off used in the manuscript.

Author Response

The authors sincerely appreciate the reviewers’ time and effort in evaluating our manuscript. Their constructive comments and suggestions have significantly improved the quality of our work. Below, we provide a point-by-point response to all the questions and comments raised by the reviewers.

   Reviewer 2:

  1. The novelty of the study should be addressed. Specifically, how does this work add to or differ from prior studies?

R/ The authors are grateful for this comment. The novelty of our study was to provide real-world evidence of the long-term prognostic value of p-tau217 in memory clinic patients. We addressed this issue in the introduction (lines 59-62 and 69-71).

2. More detail is needed on the clinical criteria used to diagnose non-AD dementias. For example, please clarify whether a cognitively normal control group was considered or used for reference.

R/ As mentioned in lines 385 to 390, “since the clinical use of blood biomarkers is currently intended for the evaluation of symptomatic individuals rather than cognitively unimpaired ones, we included only individuals with objective memory impairment in our study. Consequently, our study population is representative of those who would require a blood test to evaluate the underlying cause of their cognitive deterioration in a memory clinic”. In fact, the inclusion criterion was the presentation of suspected cognitive dysfunction for which the neurologist at the memory clinic requested CSF analysis (lines 86-87). As for ethical issues we could not collect CSF from healthy subjects, we did not include healthy controls in our study. However, our study population consisted of patients who attended a memory clinic; therefore, our study represents a more realistic application of plasma biomarkers in daily clinical practice compared with having a study population composed of healthy control subjects and AD patients. The vast majority of those who attend a memory clinic are those with MCI who may progress or not to AD or other types of dementia.

3. The association between eGFR and plasma p-tau217 is notable. The authors may consider discussing whether adjusting p-tau217 for renal function could further enhance diagnostic accuracy or aid clinical interpretation.

R/ Thank you very much for the comment. We included the following statement to the discussion: “Nevertheless, this association should be carefully considered, particularly in patients with severe kidney disease.” (lines 334-336).

4. The abstract could be improved by including numerical values (e.g., AUC) when describing the performance of p-tau217.

R/ Thank you for the comment. We agree that including more numerical data would enhance the abstract; however, the journal restricts abstracts to a maximum of 200 words. To comply, we added key numerical information while removing less relevant details to stay within the word limit (Lines 20-26).

5. Too many tables in the main text. The authors may consider moving some tables to supporting information.

R/ The Table 2 was moved to the supplementary material and the table numbers were modified accordingly.

6. Please clarify the cut-off used in the manuscript.

R/ We apologize for any confusion. It is unclear which cutoffs the reviewer is referring to. If the question concerns the cutoffs used for CSF AD biomarkers, these are provided in lines 113–115 (Sample Analysis section). If the reviewer is referring to the cutoffs determined for plasma p-tau217, they are reported in lines 272–278 and in Table 5 ("Binary and three-range reference for CSF Aβ42/40 status") under the characteristics row. If you have any further questions, we remain at your disposal to clarify them.
